# Cinnamaldehyde Could Reduce the Accumulation of Diarrhetic Shellfish Toxins in the Digestive Gland of the Mussel *Perna viridis* under Laboratory Conditions

**DOI:** 10.3390/md19020063

**Published:** 2021-01-27

**Authors:** Guo-Fang Duan, Yang Liu, Li-Na Zhang, Hong-Ye Li, Jie-Sheng Liu, Wei-Dong Yang

**Affiliations:** Key Laboratory of Aquatic Eutrophication and Control of Harmful Algal Blooms of Guangdong Higher Education Institute, College of Life Science and Technology, Jinan University, Guangzhou 510632, China; dgf1633071004@stu2016.jnu.edu.cn (G.-F.D.); ly2019@stu2019.jnu.edu.cn (Y.L.); rena2018@stu2018.jnu.edu.cn (L.-N.Z.); thyli@jnu.edu.cn (H.-Y.L.); tjsliu@jnu.edu.cn (J.-S.L.)

**Keywords:** okadaic acid, *Perna viridis*, cinnamaldehyde, CYP3A4

## Abstract

Diarrhetic shellfish toxins (DSTs), some of the most important phycotoxins, are distributed almost all over the world, posing a great threat to human health through the food chain. Therefore, it is of great significance to find effective methods to reduce toxin accumulation in shellfish. In this paper, we observed the effects of four phytochemicals including cinnamaldehyde (CA), quercetin, oridonin and allicin on the accumulation of DSTs in the digestive gland of *Perna viridis* after exposure to the DSTs-producing *Prorocentrum lima*. We found that, among the four phytochemicals, CA could effectively decrease the accumulation of DSTs (okadaic acid-eq) in the digestive gland of *P. viridis*. Further evidence demonstrated that CA could reduce the histological alterations of the digestive gland of a mussel caused by DSTs. RT-qPCR showed that CA could suppress the CYP3A4 induction by DSTs, suggesting that the DSTs’ decrease induced by CA might be related to the inhibition of CYP3A4 transcription induction. However, further studies on the underlying mechanism, optimal treatment time, ecological safety and cost should be addressed before cinnamaldehyde is used to decrease the accumulation of DSTs in field.

## 1. Introduction

One of the prominent hazards caused by harmful algae is phycotoxins. Phycotoxins can be concentrated in bivalve mollusks, mainly in the digestive gland, without causing significant adverse effects on filter-feeding bivalves [1,2,3]. However, the contaminated bivalve mollusks with phycotoxins are dangerous to human beings; in some cases, they are fatal. When the bivalves contaminated by phycotoxins are consumed by humans, severe intoxication may occur to the consumers [4]. In terms of their resultant syndrome, the phycotoxins are often grouped into five categories, including paralytic shellfish toxins (PSTs), diarrhetic shellfish toxins (DSTs), amnesic shellfish toxins (ASTs), neurotoxins and azaspiracids [5]. Among them, DSTs and PSTs are found almost all over of the world, posing a great threat to consumer’s health and safety [6,7]. Therefore, it is of great significance to find effective methods to reduce toxin accumulation in shellfish.

To date, a variety of physical, chemical and other methods have been proposed to eliminate the shellfish toxins from bivalve mollusks [6,8,9]. However, none of these methods has been proved to be effective. Some of them are difficult to use in practice, especially in live shellfish, and others are time-consuming and costly with many uncertain factors. Recently, Peña-Llopis et al. found that N-acetylcysteine boosted organophosphorus pesticide detoxification through increasing the biotransformation by glutathione S-transferase (GST) activity induction, which opened a window for screening chemicals that can eliminate toxins from shellfish [10]. It has been reported that shellfish toxins undergo a series of complex metabolic and detoxification processes in bivalves [9,11,12]. Therefore, we speculate that some chemicals that can activate the metabolic detoxification process could accelerate the elimination of toxins from shellfish. 

So far, a variety of phytochemicals that can effectively activate key enzymes of detoxification metabolism have been found [13,14]. These findings provide us with the convenience to screen substances that can accelerate the elimination of toxins from shellfish tissues due to their many advantages, including relative inexpensiveness, abundant sources, lower toxicity, and biochemical specificity [15]. However, it is of note that there are great species differences in the detoxification of xenobiotic compounds between human and bivalves, even among different bivalve species.

As mentioned above, DSTs in shellfish are a worldwide problem for food safety and public health [16]. They are typical polyether phycotoxins, including okadaic acid (OA), dinophysistoxins (DTXs) and other derivatives, which can be produced by genera of *Prorocentrum* and *Dynophysis* [17]. Many studies have shown that the accumulated DSTs can be metabolized by bivalves [2]. The involvement of detoxifying enzymes, hydrolyzing enzymes, antioxidant enzymes and ATP-binding cassette (ABC) transporters has been suggested [11,18], such as cytochrome 450 (CYP450), GSTs, P-glycoprotein (P-gp) or multidrug resistance-associated proteins (MRP) [19,20,21,22,23,24]. Studies have shown that CYP3A, especially CYP3A4, was involved in the metabolism and detoxification of DSTs in human cell lines [25,26,27]. Wei et al. reported that CYP3A4 might also play an important role in DSTs metabolism in bivalves [24]. 

In this study, we would like to find some phytochemicals that can eliminate DSTs in shellfish. The potential phytochemicals should be edible, common, efficient and convenient to practice. Therefore, we selected several accessible food additives from phytochemicals, which have been proved to exert health beneficial effects in humans. Cinnamaldehyde (CA), an α, β-unsaturated aldehyde found in cinnamon bark, is often used as a natural flavor and natural flavorant in the kitchen and industry [28]. It has antimicrobial, antiviral and anti-inflammation activities [15,29]. Allicin (ALL) is an organosulfur compound mainly found in garlic, with antimicrobial, anti-oxidative and immunomodulatory activities [30]. Oridonin (ORI), the major active ingredient of the traditional Chinese medicinal herb *Rabdosia rubescens*, has anti-oxidative and anti-tumor capacities and anti-inflammation activities [31,32]. Quercetin (QUR) is a natural flavonoid found abundantly in almost all edible vegetables and fruits, having anti-oxidative, anti-inflammatory, and antiviral activities [33,34]. Bivalve mussel *Perna viridis* is an economically important mussel, often used as a biological model to investigate physiological responses to the transitional environments due to its high tolerance to a range of contaminants [35,36]. The toxic microalgal species *P. lima*, as a reliable producer of DSTs, has been extensively used in toxicological studies concerning DSTs [37,38]. So, we employed the mussel *P. viridis* as a model, and used the microalga *P. lima* as a DST source in our study. Firstly, we analyzed the effects of the four substances CA, ALL, ORI, and QUR on the accumulation of DSTs in the mussel *P. viridis* after exposure to *P. lima*. In addition, the expression of several enzymes concerning xenobiotic metabolism including CYP3A4, CYP3A1, and the histological alterations of the digestive gland were observed.

## 2. Results

### 2.1. CA Can Significantly Reduce the DST Accumulation in Mussel P. viridis

As demonstrated in Figure 1A, the DST contents in *P. lima*-exposed groups were remarkably higher than the control counterparts (*p* < 0.05). The *P. lima*-exposed mussels with CA (group: *P. lima* + CA) accumulated less DSTs than those exposed only to *P. lima* after 12 h (*p* < 0.05). However, there is no significant difference in DST content between the *P. lima*-exposed mussels without any phytochemicals and ones with ALL, ORI or QUR both at 6 and 12 h (*p* > 0.05). These results suggest that, among the four phytochemicals assayed, only CA significantly reduced the DST accumulation in the mussel *P. viridis*. 

### 2.2. Effect of CA on the DST Accumulation in the Digestive Gland Is Concentration Dependent

The effects of different concentrations of CA on the DST accumulation in the digestive gland after exposure to *P. lima* are shown in Figure 1B. CA at 1 μM could not reduce DST accumulation in the digestive gland of the *P. lima*-exposed mussels. However, when the concentrations of CA were 15, 30, 60, and 120 μM, the contents of DSTs in the *P. lima*-exposed mussels with CA were significantly lower than those exposed only to *P. lima* (*p* < 0.01). The contents of DSTs in 15, 30, 60 and 120 μM of CA treatment groups were only 60.3, 74.7, 37.7 and 43.0 ng g^−1^, respectively. It looks like that the effect of CA on DST accumulation in the digestive gland is concentration dependent, and 60 μM seems to be the most appropriate concentration of CA.

### 2.3. CA Could Reduce the DST Accumulation in the Digestive Gland but Not in the Gills of P. viridis

Figure 2 shows the DST contents in the digestive gland and in the gills of the *P. lima*-exposed *P. viridis* at different times after the addition of CA (60 μM). The DST contents in the *P. lima*-exposed mussels were higher than the control counterparts both at 12 and 48 h (*p* < 0.05). In digestive glands, the contents of DSTs in the presence of CA were significantly lower than those of the *P. lima*-exposed mussels without CA at 12 and 48 h (*p* < 0.05). However, there was no significant difference in DST content in the gills between the *P. lima*-exposed mussels with and without CA.

To explore the fate of DSTs eliminated from the digestive gland, we measured the content of DSTs in the cultures at 12 and 48 h after the addition of CA. As in Figure 2, the concentrations of DSTs in seawater from the *P. lima*-exposed groups with CA (*P. lima* + CA group) and without CA (*P. lima* group) were higher than those of the control (*p* < 0.05). The DST concentration in seawater from the *P. lima*-exposed groups with CA (*P. lima* + CA group) was significantly higher than that from the *P. lima*-exposed group without CA (*p* < 0.05).

### 2.4. CA Could Alleviate the Histological Alterations of the Digestive Gland of Mussels Caused by DSTs

Digestive gland tissues 12, 24 and 48 h after the addition of CA was used to evaluate the histological alterations after being exposed to *P. lima* with CA and without CA. As in Figure 3, the digestive gland tissue of the *P. lima*-exposed mussel exhibited some structural changes. In the control, the columnar epithelial cells of digestive diverticulum arranged neatly, and the digestive tubules connected with each other through connective tissue; no atrophy of epithelial cells, infiltration or necrosis of hemocyte were observed. Meanwhile, no significant changes in the structure were observed in the CA group (Figure 3B). However, in the *P. lima*-exposed mussel, as shown in Figure 3C, the structure of digestive gland tissue presented distinct changes. At 12 h, the epithelial cells were severely atrophied and the digestive tubules disappeared. Meanwhile, the connective tissues were broken, and the digestive diverticulum formed vacuoles. At 24 h, the epithelial cells were highly degraded and moderately atrophied, and the columnar epithelial cells were arranged disorderly. There was evidence of aggregation and infiltration of blood cells. At 48 h, the epithelial cells were severely atrophied, accompanied by partial epithelial cell decomposition, the disappearance of digestive cells, destruction of digestive tubules, and the deformation of digestive diverticulum. 

In contrast, the damages of digestive glands in the *P. lima* + CA group were significantly less than the *P. lima* group. The epithelial cells showed moderate atrophy at 12 h and slight atrophy at 48 h after the addition of CA, and no infiltration of blood cells or decomposition of epithelial cells was observed (Figure 3D). 

### 2.5. Changes in the Expression of Genes Concerning Metabolism

Figure 4 provides the relative expression of some genes concerning metabolism and detoxification in the digestive gland of *P. lima*-exposed mussels with or without CA. It can be seen that the addition of CA had some effects on the expression of CYP3A4 and CYP3A1 in the digestive gland of the *P. lima*-exposed mussels. 

The level of CYP3A4 mRNA in the *P. lima*-exposed mussels with CA (*P. lima* + CA group) was significantly lower than that without CA (*P. lima* group) both at 12 and 48 h, though the expression of CYP3A4 mRNA was significantly up-regulated in the *P. lima*-exposed mussels without CA compared with the control at 48 h (*p* < 0.05). In contrast, the level of CYP3A1 mRNA in *P. lima*-exposed mussels with CA (*P. lima* + CA group) was significantly higher than that without CA (*P. lima* group) at 12 h. These outcomes suggest that CA might change the metabolism process of DSTs.

## 3. Discussion

At present, a variety of shellfish toxin detoxification methods have been developed, including physical methods, biological methods, and chemical methods [39,40,41]. However, no method has been proved to be effective in the detoxification of live shellfish. Food biotechnology has discovered a variety of natural substances that can effectively activate key metabolism and detoxification enzymes in humans, providing an opportunity for us to find safe and effective substances for DST detoxification [13]. In this paper, we prospected the effects of four phytochemicals, CA, ALL, ORI and QUR, on DST accumulation after exposure to *P. lima* cells in the mussel *P. viridis*. We found that, among the four natural products assayed, only CA significantly decreased the accumulation of DSTs in the digestive gland tissue of the *P. lima*-exposed mussel. The DST contents in the control were not zero as shown in Figure 1A, which could be due to the matrix effect [20]. Further evidence showed that CA could alleviate the damages of the digestive gland induced by *P. lima* cells, corroborating the detoxification of CA on DSTs in mussels. After being ingested, DSTs mainly accumulated in the digestive gland and in the gills of bivalves, while other tissues such as mantle and adductor tissue only comprised a very low concentration of toxins [42].

To further learn the fate of DSTs accumulated in the digestive gland tissue, concentrations of DSTs in gill tissue and seawater from the culture were evaluated. We found that the addition of CA increased the content of DSTs in seawater, but had no distinct effect on DST content in the gill tissue. In this study, the content of DSTs was detected by ELISA assay. This method is not able to discriminate the different types of DSTs, including OA, DTX1, diol esters, DTX3 and DTX4-DTX5-type compounds. Meanwhile, some of these compounds present a limited cross-reactivity in the ELISA assay used. Therefore, we could not distinguish possible changes in DST biotransformation caused by the addition of CA. Nevertheless, the decrease in OA content in the digestive gland and the increase in OA content in the culture medium, together with no significant change in OA content in the gill tissue, gave some evidence that CA might not change the distribution of DSTs in mussels but facilitate the elimination of DSTs from the digestive gland.

Detoxification of bivalve molluscs is a complex process concerning cytochromes P450 and other enzymes and transporters. Studies have shown that CYP3A, such as CYP3A4 and CYP3A1, were involved in the metabolism and detoxification of DSTs in mammalian cell lines [25,26,27]. Recently, based on the finding that the content of DSTs was significantly decreased in mussels when the activity of CYP3A4 was inhibited by ketoconazole, Wei et al. proposed that CYP3A4 might play an important role in DST metabolism in bivalves. So, we assayed the changes in CYP3A4 and CYP3A1 transcripts [24]. We found that the expression of CYP3A4 in the digestive gland of the *P. lima*-exposed mussel was significantly up-regulated at 48 h compared with the control counterparts. However, the relative expression level of CYP3A4 in the *P. lima*-exposed mussel with CA (*P. lima* + CA group) was significantly lower than that without CA (*P. lima* group) both at 12 and 48 h, indicating that CA reduced the expression of CYP3A4 mRNA in the digestive gland induced by DSTs. It is likely that the DSTs’ decrease in the digestive gland in the presence of CA might be due to the inhibition of CYP3A4 induction. Additionally, the CYP3A1 transcript displayed an alteration at 12 h after the addition of CA, suggesting that CA might also exert an effect on CYP3A1 mRNA and that CYP3A1 could be involved in the elimination of DSTs. 

Many studies have shown that the accumulated DSTs can be metabolized by bivalves [2]. CYP3A4 is transcriptionally regulated by several nuclear receptors, such as the pregnane X receptor (PXR) and constitutive androstane receptor (CAR) [43]. Nuclear receptors, such as the PXR, CAR, and vitamin D (1,25-dihydroxyvitamin D3) receptor (VDR), can bind to the retinoid X receptor (RXR) to form a heterodimer, thereby regulating the transcription of CYP3A genes [44]. AhR, a key regulator of the metabolism of exogenous chemicals in multiple species, might also be involved in modulation of CYP genes [45,46]. Therefore, we proposed that CA might down-regulate the expression of CYP3A4 by modulating the expression of some nuclear receptors. However, further studies are warranted to explore the potential mechanism of CA to reduce toxin accumulation in bivalves. 

Histopathological observation is the most direct way to understand the physiological alteration in organism after exposure to toxic substances [38]. The digestive gland is the main organ of metabolic regulation, immune defense mechanism, and homeostatic regulation in mollusks. Histomorphology can directly reflect the effects of toxin accumulation on shellfish health [47]. Therefore, we observed the morphological changes of the digestive gland in the DSTs-exposed mussels. The results demonstrate that the *P. lima* cells caused severe damages to the digestive gland tissue of *P. viridis*. In line with the observation by Neves et al. in mussels and de Jesús Romero-Geraldo et al. in oyster, these impairments of mussels mainly manifested in the tubular atrophy of digestive gland diverticula, the hemocyte infiltration, the decrease in digestive cells, and the atrophy and decomposition of epithelial cells, which might weaken the ability to digest the granular matter, the promotion of pseudofeces production, and the increase in mucus secretion of the *P. lima*-exposed mussel [38,48]. However, the hemocyte infiltration and the decomposition of epithelial cells were not observed in digestive gland tissue after the addition of CA, though some atrophy of the epithelial cells was still observed, indicating that CA could reduce the damage of DSTs to mussels.

In agreement with the previous observation, some behaviors of the *P. lima*-exposed mussels changed, as featured in the partial shell-valve closure, the generation of pseudofeces, and the production of mucus, etc. [37]. Correspondingly, the culture water became turbid and the mucus secretion increased significantly on the second day after exposure to *P. lima*. However, in the presence of CA (*P. lima* + CA group), the culture water was obviously cleaner than that of the *P. lima*-exposed mussel without CA (*P. lima* group). This improvement gave further evidence that CA relieved damages of DSTs to the digestive gland.

It should be pointed out that, in this paper, we only detected the content of DSTs in the digestive gland of mussels at two moments after the exposure to *P. lima* and CA was only added once at 2 h after exposure. Therefore, time effect tests need to be conducted in the future to find the optimal treatment time for its field application. In addition, CA has been widely used in kitchen and industry as a natural flavor and natural flavorant, but its ecological safety and economic cost remain unclear, which should be assessed before practical application. Nevertheless, our finding might provide a new way or idea for the depuration of shellfish toxins in bivalves.

## 4. Materials and Methods

### 4.1. Materials and Animals

Given the potential effect of abiotic factors (e.g., pH, salinity) and biotic factors (such as interactions between host and microbe) on DST accumulation in bivalve species [49], microalgal species and mussel were cultured with natural seawater filtrated through 0.22-μm filters.

The DSTs-producing dinoflagellate *P. lima* (CCMP 2579) was purchased from the National Center for Marine Algae and Microbiota (NCMA) in the USA, which have been proved to produce OA and DTX1 [17,50]. The chlorophyte *Tetraselmis subcordiformis* was kindly provided by the Institute of Aquatic Biology, Chinese Academy of Sciences (Wuhan, China). The two strains were cultured in f/2-Si medium, which were filter sterilized through 0.22 µm filters. The cultures were grown at 20 ± 1 °C in an artificial climate incubator (60 µmol m^−2^ s^−1^, 12/12 h light/dark cycle).

The mussel *P. viridis* (9–10 cm) was purchased from the Guangzhou Huangsha Seafood Market, which collected from the Zhanjiang sea area in Guangdong province, China. After removing the mud, deposits, and barnacle parasites from the surface of the shell, the *P. viridis* individuals were raised in several aerated aquariums (300 mm × 450 mm × 300 mm) under controlled conditions (12 h light/12 h dark cycle, 17.5 ± 1 °C) for at least 7 days prior to experiments. The culture water in aquariums was completely renewed daily at a fixed time, and animals were fed with 1 × 10^7^ cell/L cultured *T. subcordiformis*. The individuals that have grown well and with strong foot filaments were selected for experiments.

ALL (purity ≥ 98%) and QUR (purity ≥ 98.5%) were purchased from Macleans (Shanghai, China), whereas CA (purity ≥ 99.5%) and ORI (purity ≥ 98%) were purchased from Aladdin (Shanghai, China). The four substances were all dissolved in methanol. 

### 4.2. Experimental Design

Given the findings that DSTs mainly concentrate in the digestive gland of mussel *P. viridis* [24,42], we focused on the DST content in the digestive gland tissue of the *P. lima*-exposed *P. viridis* after different phytochemicals or different phytochemical concentration or different phytochemical exposure time treatments. We had designed our study in three steps as shown in Figure 5.

Step I: Prospection of the four phytochemicals. A total of 216 acclimatized mussel *P. viridis* individuals were randomly assigned to 6 aquariums (36 mussels in each aquarium) with the same specification. The mussels in the control group were only fed with 1 × 10^7^ cell/L *T. subcordiformis*, while mussels in other aquariums (experimental group) were fed with both 1 × 10^7^ cell/L *T. subcordiformis* and 2 × 10^6^ cell/L *P. lima*. After feeding microalgae for 2 h, phytochemicals ALL, QUR, CA or ORI (dissolved in 150 μL methanol) were added to the aquariums in experimental groups, respectively, with a final concentration of 20 μM, whereas isovolumetric solvent was added to the aquarium in the control group. The digestive gland tissues were collected at 6 and 12 h from 36 (18 at each time point) mussels in each aquarium after the addition of phytochemicals for OA detection (at 8 and 14 h after *P. lima* exposure). In order to diminish individual differences, tissues from six mussel individuals within the same treatment were pooled together as one sample. Finally, three biological repetitions were obtained at each sampling time point in each group and stored at −20 °C for DST detection.

Step II: Concentration dependence of the most appropriate phytochemical. A total of 108 acclimatized mussel *P. viridis* individuals were employed, which randomly assigned to 6 aquariums (18 mussels in each aquarium) with the same specification. All the mussels were only fed with both 1 × 10^7^ cell/L *T. subcordiformis* and 2 × 10^6^ cell/L *P. lima*. After feeding microalgae for 2 h, CA (dissolved in 150 μL methanol) was added to the aquariums with different final concentrations (0, 1, 15, 30, 60, and 120 μM, respectively). All the mussels were slaughtered for digestive gland tissue collection at 48 h after the addition of CA or solvent. In order to diminish individual differences, tissues from six mussel individuals within the same treatment were pooled together as one sample. So, three biological repetitions were obtained in each group.

Step III: Time-dependence of the most appropriate phytochemical. A total of 216 acclimatized mussel *P. viridis* individuals were randomly assigned to 4 aquariums (54 mussels in each aquarium) with the same specification. The mussels in the aquarium of the control group and CA group were only fed with 1 × 10^7^ cell/L *T. subcordiformis*, while mussels in the aquarium of the *P. lima* group and *P. lima* + CA group were fed with both 1 × 10^7^ cell/L *T. subcordiformis* and 2 × 10^6^ cell/L *P. lima*. After feeding microalgae for 2 h, CA were added to the aquarium in the CA group and *P. lima* + CA group with a final concentration of 60 μM, whereas isovolumetric solvent was added to the aquarium of the control group and *P. lima* group. At 12 h, 48 h after the addition of CA, eighteen individuals were randomly sampled at each fixed time point in each group, respectively. Gills or digestive glands at each time point within the same treatment were excised and stored until subsequent OA detection (stored in −20 °C refrigerator) and RNA extraction (stored in −80 °C refrigerator). To diminish individual differences, tissues from six mussel individuals within the same treatment at each time point were pooled together as one sample as did in Step I and II. Therefore, three biological repetitions were obtained in each group at each time point for OA detection and RNA extractions. In addition, seawater from the cultures were sampled at 12 and 48 h after the addition of CA and stored at 4 °C for OA detection. In the case of the histological examination, digestive gland tissues were sampled at 12 h, 24 h, and 48 h after addition of CA. The samples were fixed in 4% paraformaldehyde fixative for 48 h for paraffin section.

### 4.3. DSTs Quantitative Detection

DSTs in tissue samples were extracted and quantified according to the manufacturer’s instructions of Okadaic Acid (DSP) ELISA Test Kit (Abraxis, Warminster, PA, USA) as described in our previous paper [24]. Briefly, 1 g of fresh tissue homogenate was mixed with 6 mL 80% methanol solution. After being centrifuged at 3000× *g* at 4 °C for 10 min, the supernatant was collected. Then, another 2 mL 80% methanol solution was added to the tissue residue for re-extraction. The supernatant was combined and filtered with a 0.45 μm filter (Millex HV, Millipore, Darmstadt, Germany). The quantification of DSTs was performed using an Okadaic Acid (DSP) ELISA Test Kit (Abraxis, Warminster, PA, USA) according to the manufacturer’s instructions, which recognizes OA and other DSTs with varying degrees (OA, 100%; DTX-1, 50%; DTX-2, 50%). The content of DSTs was expressed as ng OA eq/g. 

DSTs in cultures were concentrated by using an Oasis HLB 6 cc (500 mg) LP Extraction Cartridge (Waters Ltd., Worcester, MA, USA) filter column as described by Fang et al. with slight modifications [51]. A Supelco Visiprep 12 tube anti-contamination (DL) solid phase extraction unit was used. Firstly, an HLB filter column was activated by methanol, and washed with distilled water, then 30 mL of seawater sample was added. Thereafter, the column was eluted successively with 3 mL methanol and 2 mL 30% methanol. Finally, the eluent was collected, and dried with nitrogen, then dissolved in 500 μL of methanol for DSTs detection. DST quantification was also carried out by using an Okadaic Acid (DSP) ELISA Test Kit (Abraxis, Warminster, PA, USA) according to the manufacturer’s instructions. 

### 4.4. Quantitative Polymerase Chain Reaction (qPCR)

Total RNA was extracted by Total RNA Kit I (50) R6934-01 (Omega, Norcross, GA, USA). RNA integrity was tested by denaturing agarose gel electrophoresis, and RNA concentration was measured by using NanoDrop 2000/2000c Spectrophotometers (Implen, Munich, Germany) at 260 nm. Approximately 1 μg of total RNA was reverse transcribed into cDNA by HiScript^®^ II Q RT SuperMix for qPCR (+gDNA wiper) (R223-01, Vazyme, Nanjing, China) kit according to the manufacturer’s instructions.

The expression of five candidate genes—elongation factor 1 alpha (EF1α), ribosomal protein L3 (RPL3), ribosomal protein L13-like (RPL13), ribosomal protein L37 (RPL37) and ubiquitin A-52 (UBA52)—was examined by using geNorm, NormFinder and BestKepper. The most stable expressed genes RPL3 and RPL13 were used as reference genes to normalize the expression of target genes including CYP3A4 and CYP3A1. All the primers were designed based on the sequences of *Mytilus edulis* by using Primer 5.0 (Table 1). 

PCR was performed on a CFX96 Real-Time PCR System (Bio-Rad, Hercules, CA, USA) using AceQ^®^ qPCR SYBR^®^ Green Master Mix (Vazyme, Nanjing, China) with the following profile: 95 °C for 5 min, 40 cycles of 95 °C for 10 s, 60 °C for 30 s. The PCR specificity was evaluated by melting curve analyses from 65 to 95 °C raising by 0.5 °C at 5 s. The reaction mixture (20 μL) consisted of 10 µL of AceQ^®^ qPCR SYBR^®^ Green Master Mix, 7.6 µL of H_2_O, 0.2 µL of each forward and reverse primers (10 µM) and 2 µL of first-strand cDNA template from 1 μg RNA. No template control (NTC) amplifications were also performed on equivalent double-distilled water to check the absence of contaminant. Inter-run calibrators (IRC) were set for eliminating errors of two plates running data. Amplification efficiency for all the genes assayed should range from 98% to 105%, and the correlation coefficient should be more than 0.99.

The comparative C_q_ method was used to analyze the relative expression level of genes as described by Derveaux et al., where multiple reference genes and inter-run calibration algorithms were considered [52].

### 4.5. Histological Examination

Digestive gland tissues sampled at 12, 24, and 48 h in Step III were used for histological examination. Histological preparations were performed as described in previous reports with some modifications [38,47]. Briefly, as far as possible, a relatively complete digestive gland tissue was fixed in 4% paraformaldehyde solution for 48 h; subsequently, the tissue was dehydrated and embedded in paraffin according to the routine process of paraffin embedding. The embedded tissue was sectioned at 4 μm thickness with microtome (Leica RM2235, Lecia Microsystems Nussloch Gmbh, Heidelberger, Germany). After being dried in an incubator at 37 °C overnight, the obtained paraffin-embedded section was deparaffinized in xylene, rehydrated in gradient ethanol solutions, and then stained with hematoxylin and eosin. The stained slice was sealed with neutral balsam, and then the images were taken at 40× magnification with Pannoramic MIDI Slide Scanner (3DHISTECH, Budapest, Hungary) and analyzed by CaseViewer software (3DHISTECH, Budapest, Hungary). 

### 4.6. Statistical Analyses

Statistical analyses were performed by using software SPSS Statistics 22.0. All data were expressed as mean ± SD. In Steps I and III, Fisher’s protected multiple comparisons Least Significant Difference (LSD) test was employed to compare differences between different treatment groups, with significant differences at *p* < 0.05. In Step II, a *t*-test was used to detect differences in DST content between the control (with 0 μM CA) and experimental groups.

## 5. Conclusions

Among the four phytochemicals—cinnamaldehyde, quercetin, oridonin and allicin—only cinnamaldehyde could reduce the accumulation of DSTs in the digestive gland of mussels. Furthermore, cinnamaldehyde relieved damages of DSTs to the mussel corroborating its detoxification on DSTs in mussels. After exposure to *P. lima*, the expression of CYP3A4 was over-expressed in the digestive gland of the mussels, but CA inhibited this overexpression. Considering the decrease in DST accumulation when CYP3A4 activity was inhibited, we proposed that the decrease induced by CA might be due to the inhibition of CYP3A4 induction. However, further studies are warranted to explore the potential mechanism of CA to reduce toxin accumulation in bivalves.

## Figures and Tables

**Figure 1 marinedrugs-19-00063-f001:**
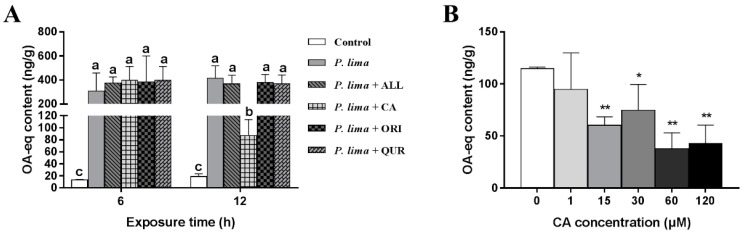
Changes in diarrhetic shellfish toxin (DST) content (ng okadaic acid (OA) eq/g) in the digestive gland of the *Prorocentrum lima*-exposed mussels after the addition of phytochemicals. (**A**) DST contents in digestive glands of the *P. lima*-exposed mussels in the presence of different phytochemicals (20 μM). Bars of respective treatment followed by the same letter are not significantly different at *p* < 0.05 (Fisher’s protected multiple comparisons LSD test). (**B**) DST contents in digestive glands of the *P. lima*-exposed mussels in the presence of different concentrations of CA. Significant differences compared to control (with 0 μM CA) are represented by asterisks (*t*-test, * *p* < 0.05; ** *p* < 0.01). All data are expressed as mean ± SD (*n* = 3). Abbr. ALL, allicin; CA, cinnamaldehyde; ORI, oridonin; QUR, quercetin.

**Figure 2 marinedrugs-19-00063-f002:**
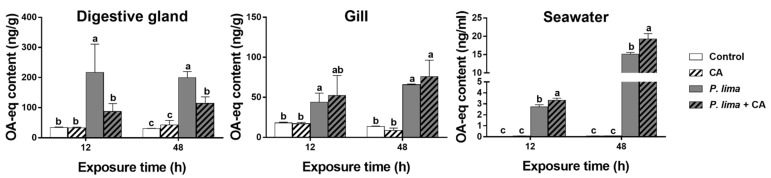
DST contents (ng OA eq/g) in the digestive glands and gills of the *Prorocentrum lima*-exposed mussels and in seawater (culture solution) at 12 and 48 h after the addition of cinnamaldehyde (CA). Data are expressed as mean ± SD (*n* = 3). Bars of respective treatment followed by the same letter are not significantly different at *p* < 0.05 (Fisher’s protected multiple comparisons LSD test). Control, *Tetraselmis subcordiformis* (1 × 10^7^ cells/L); CA, *T. subcordiformis* (1 × 10^7^ cells/L) + CA (60 μM); *P. lima*, *T. subcordiformis* (1 × 10^7^ cells/L) + *P. lima* (2 × 10^6^ cells/L); *P. lima* + CA, *T. subcordiformis* (1 × 10^7^ cells/L) + *P. lima* (2 × 10^6^ cells/L) + CA (60 μM).

**Figure 3 marinedrugs-19-00063-f003:**
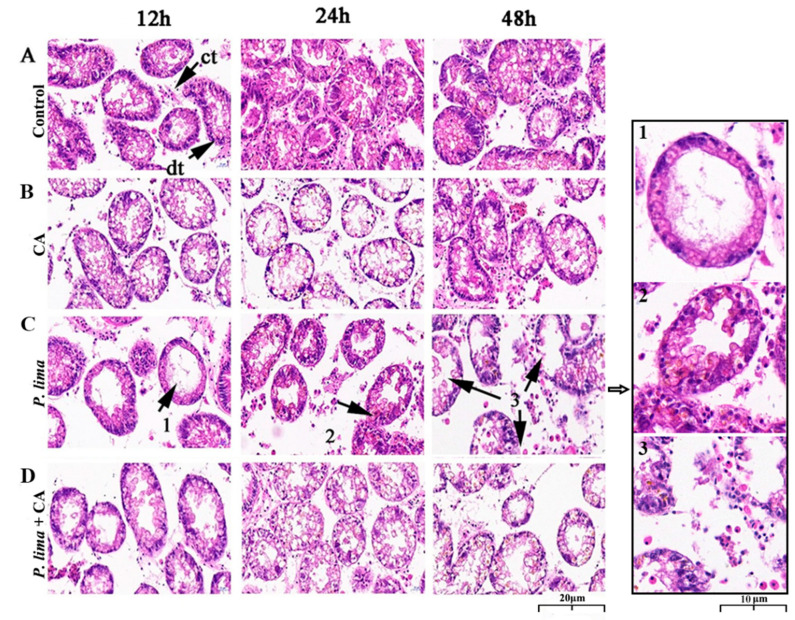
Digestive gland tissue sections of the *Prorocentrum lima*-exposed mussel *Perna viridis* (HE staining, ×400) at 12, 24 and 48 h after the addition of cinnamaldehyde. (**A**) Control, *Tetraselmis subcordiformis* (1 × 10^7^ cells/L); (**B**) CA, *T. subcordiformis* (1 × 10^7^ cells/L) + CA (60 μM); (**C**) *P. lima*, *T. subcordiformis* (1 × 10^7^ cells/L) + *P. lima* (2 × 10^6^ cells/L); (**D**) *P. lima* + CA, *T. subcordiformis* (1 × 10^7^ cells/L) + *P. lima* (2 × 10^6^ cells/L) + CA (60 μM). Abbr. CA, cinnamaldehyde; dt, digestive tubule; ct, connective tissues. Mark 1, severely atrophied epithelial cells; Mark 2, disorderly arranged columnar epithelial cells and hemocyte infiltration; Mark 3, epithelial cells decomposed, digestive tubules were destroyed, and digestive gland diverticula were deformed.

**Figure 4 marinedrugs-19-00063-f004:**
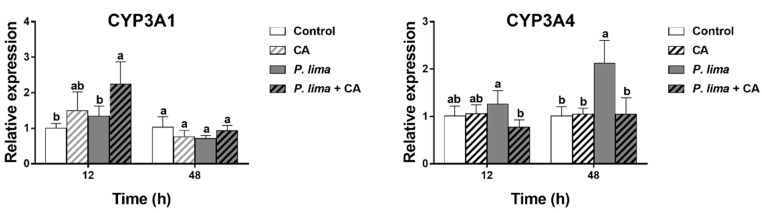
Changes in the expression of CYP3A1 and CYP3A4 in the digestive glands of the *Prorocentrum lima*-exposed mussels revealed by RT-qPCR. The values are expressed as mean ± SD. Bars of respective treatment followed by the same letter are not significantly different at *p* < 0.05 (Fisher’s protected multiple comparisons LSD test). Control, *Tetraselmis subcordiformis* (1 × 10^7^ cells/L); CA, *T. subcordiformis* (1 × 10^7^ cells/L) + CA (60 μM); *P. lima*, *T. subcordiformis* (1 × 10^7^ cells/L) + *P. lima* (2 × 10^6^ cells/L); *P. lima* + CA, *T. subcordiformis* (1 × 10^7^ cells/L) + *P. lima* (2 × 10^6^ cells/L) + CA (60 μM). Abbr. CA, cinnamaldehyde.

**Figure 5 marinedrugs-19-00063-f005:**
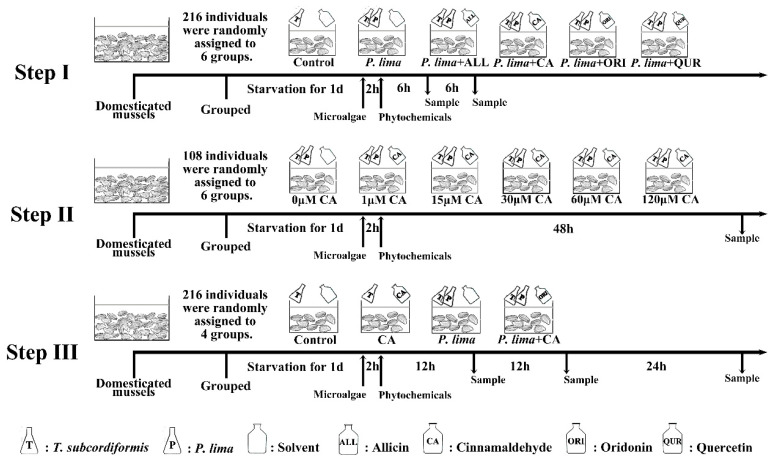
Experimental design. **Step I**: Prospection of the four phytochemicals (ALL, CA, ORI and QUR); **Step II**: Concentration dependence of the most appropriate phytochemical; **Step III**: Time dependence of the most appropriate phytochemical. The mussels were adapted to laboratory conditions for 7 days before being used in experiments. Abbr. ALL, allicin; CA, cinnamaldehyde; ORI, oridonin; QUR, quercetin; *P. lima*, *Prorocentrum lima*; *T. subcordiformis, Tetraselmis subcordiformis.*

**Table 1 marinedrugs-19-00063-t001:** Primers for RT-qPCR.

Gene Name	Primer Sequence (5′–3′)	Amplicon Size (bp)
EF1-Fα	F: CACTCCGTCTTCCACTCCA	131
	R: CCTCTGGCATTGACTCGTG	
Tubulin-β	F: AGGAAGGAGGCTGAGAGTTGT	135
	R: TTTGGAGATGAGCAGGGTTC	
RPL3	F: GGTGGCACTATCTCCCAGAA	98
	R: GCCATCTGGACGTTACACCT	
UBA52	F: TTACATTTGGTCCTGCGTCTC	135
	R: CAGTTGGTAGCCCTTTGATGA	
RPL13	F: TAAAGACTGGCAACGCTATGT	155
	R: TCACAACTGGTCGGAGAAG	
RPL37	F: GTCGCAATAAGACGCACACGTTG	179
	R: GTGCCTCATTCGACCAGTTCCG	
CYP3A1	F: CGCTGCTGTGACGATCTGGTAG	141
	R: TCTCTGCGAATTCACCTGCAACC	
CYP3A4	F: GAGACCTTTGACCCGGAACG	99
	R: ACCAATGCAAATGCGTGGTC	

EF1α, elongation factor 1 alpha; RPL3, ribosomal protein L3; RPL13, ribosomal protein L13-like; RPL37, ribosomal protein L37; UBA52, ubiquitin A-52; CYP3A1, Cytochrome P450 3A1; CYP3A4, Cytochrome P450 3A4.

## Data Availability

The data in this study are available from the corresponding author upon request.

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
