# Peer review of "Cinnamaldehyde Could Reduce the Accumulation of Diarrhetic Shellfish Toxins in the Digestive Gland of the Mussel Perna viridis under Laboratory Conditions"

_marinedrugs, 2021, doi:10.3390/md19020063_

Round 1
Reviewer 1 Report
Dear authors,
Your paper fits the scope of the journal; it is well presented and written; The topic is highly interesting. Extensive data are provided. the conclusions support the data.
Prior to publication, please:
1- correct any typographical and grammatical errors
2- please provide HPLC data of the four phytochemical used
3- please improve the resolution of the figures and tables
4- Please provide underlying putative molecular mechanism(s)
5- please provide/specify clearly the lab conditions used, and comment about what would be the effect observed in natural conditions
6- comment about biotic and abiotic factors that may influence your data
For that purpose I encourage you to read and cite the following article:
Menaa F, Wijesinghe PAUI, Thiripuranathar G, Uzair B, Iqbal H, Khan BA, Menaa B. Ecological and Industrial Implications of Dynamic Seaweed-Associated Microbiota Interactions. Mar Drugs. 2020 Dec 14;18(12):641. doi: 10.3390/md18120641. PMID: 33327517; PMCID: PMC7764995.
The reviewer
Author Response
Point 1: correct any typographical and grammatical errors
Response 1: Thanks for your comment. We have corrected the typographical and grammatical errors.
Point 2: please provide HPLC data of the four phytochemicals used
Response 2: Thank you very much for professional reminding. We have supplemented the purity data of the four phytochemicals used in the Materials and Methods section. The purity of allicin (ALL, ≥98%) was identified by ultraviolet-visible spectroscopy (UV-VIS) method. The purity of quercetin (QUR, ≥98.5%) and Oridonin (ORI, ≥98%) was identified by high performance liquid chromatography (HPLC) method. The purity of cinnamaldehyde (CA, ≥99.5%) was identified by gas chromatography (GC) method.
Point 3: please improve the resolution of the figures and tables
Response 3: Thanks for kindly reminding. To improve the resolution, we have re-drawn the chart.
Point 4: Please provide underlying putative molecular mechanism(s)
Response 4: Thanks for your professional comment. Based on our data, CA could suppress the CYP3A4 induction by DSTs, suggesting that the DSTs decrease induced by CA might be related to the inhibition of CYP3A4 mRNA induction. Many studies have shown that the accumulated DSTs can be metabolized by bivalves (Blanco, 2018). CYP3A4 is transcriptionally regulated by several nuclear receptors such as the pregnane X receptor (PXR) and constitutive androstane receptor (CAR) (Du et al., 2017). Nuclear receptors, such as PXR, CAR, and vitamin D (1, 25-dihydroxyvitamin D3) receptor (VDR), can bind to retinoid X receptor (RXR) to form a heterodimer, thereby regulating the transcription of CYP3A genes (Chen et al., 2010). AhR, a key regulator of the metabolism of exogenous chemicals in multiple species, might also be involved in modulation of CYP genes (Esser and Rannug, 2015; Islam et al., 2017). Therefore, we proposed that CA might down-regulate the expression of CYP3A4 by suppressing the expression of some nuclear receptors. However, further studies are warranted to explore the potential mechanism of CA to reduce toxin accumulation in bivalves. Related expression has been added in the revised manuscript.
Blanco, J., Accumulation of dinophysis toxins in bivalve molluscs. Toxins (Basel) 2018, 10 (11), 453.
Chen, S.; Wang, K.; Wan, Y.J., Retinoids activate RXR/CAR-mediated pathway and induce CYP3A. Biochem. Pharm. 2010, 79(2), 270‒276.
Du, Z.H.; Xia, J.; Sun, X.C.; Li, X.N.; Zhang, C.; Zhao, H.S.; Zhu, S.Y.; Li, J.L., A novel nuclear xenobiotic receptor (AHR/PXR/CAR)-mediated mechanism of DEHP-induced cerebellar toxicity in quails (Coturnix japonica) via disrupting CYP enzyme system homeostasis. Environ. Pollut. 2017, 226, 435-443.
Esser, C.; Rannug, A., The aryl hydrocarbon receptor in barrier organ physiology, immunology, and toxicology. Pharmacol. Rev. 2015, 67(2), 259‒279.
Islam, J.; Sato, S.; Watanabe, K.; Watanabe, T.; Ardiansyah, Hirahara, K.; Aoyama, Y.; Tomita, S.; Aso, H.; Komai, M.; Shirakawa, H., Dietary tryptophan alleviates dextran sodium sulfate-induced colitis through aryl hydrocarbon receptor in mice. J. Nutr. Biochem. 2017, 42, 43‒50.
Point 5: please provide/specify clearly the lab conditions used, and comment about what would be the effect observed in natural conditions
Response 5: Thanks for your professional comment. Our experiment was performed in controlled conditions. The DSTs-producing dinoflagellate P. lima (CCMP 2579) and the chlorophyte T. subcordiformis were cultured in f/2 -Si medium, which were filter-sterilized through 0.22-µm filters. The cultures were grown at 20 ± 1 °C in an artificial climate incubator (60 µmol m-2 s-1, 12/12 h light/dark cycle). The mussel P. viridis individuals were raised in several aerated aquariums (300 mm × 450 mm × 300 mm) with filtered seawater at 17.5 ± 1 °C in an incubator. In experiment group, mussels were fed with both 1×107 cell/L T. subcordiformis and 2×106 cell/L P. lima. However, in natural conditions, mussels were exposed to complex conditions of a variety of microalgal species, environmental pollutants and other organisms (especially bacteria), which could exert some effect on the accumulation OA in bivalves. To exclude the potential biotic and abiotic factors in natural conditions, controlled conditions should be set. However, our finding might provide a new way or idea for the depuration of shellfish toxins in bivalves, though further studies should be done before practical application. Related expression has been added in the revised manuscript.
Point 6: comment about biotic and abiotic factors that may influence your data
Response 6: Thanks for your professional comment. Given the potential biotic and abiotic factors on the accumulation OA in bivalves, we performed experiment in controlled conditions. As mentioned above, the two microalgal species were cultured with f/2 -Si medium that were filter-sterilized through 0.22-µm filters. The mussel P. viridis individuals were raised in several aerated aquariums (300 mm × 450 mm × 300 mm) under controlled conditions (12 h light/ 12 h dark cycle, 17.5 ± 1 °C) for at least 7 days prior to experiments. The culture water in aquariums were completely renewed daily at a fixed time, and animals were fed with 1×107 cell/L cultured Tetraselmis subcordiformis. Only potential biotic factor that could exert effect on our data is physiological condition of the mussel. To diminish this possibility, the individuals that have grown well and with strong foot filaments were selected for experiments. The time point of adding CA to the culture could have some effect on the experimental outcome. Nevertheless, we only observed the content of DSTs in the digestive gland of mussels at two moments after exposure of P. lima, while CA was only added at 2 h after exposure. Therefore, time effect test needs to be conducted in the future. Related information has been added in the revised manuscript.
Reviewer 2 Report
Authors repot inhibition of diarrhetic shellfish toxin (DST) accumulation in mussels by cinnamaldehyde. The results are very interesting. I think the results are worthy being published in Marine Drugs.
1) In Figure 2. OA-eq content of seawater increased after 48 h. Authors should show the number of P. lima cells after 48 h.
2) DSTs were extracted with 80% MeOH from mussels. Was DTX3 extracted fully?
Author Response
Point 1: In Figure 2. OA-eq content of seawater increased after 48 h. Authors should show the number of P. lima cells after 48 h.
Response 1: Thanks for your professional advice. We failed to count the number of P. lima cells after 48 h due to the culture water becoming turbid on the second day after exposure to P. lima. However, we found that the addition of CA increased the content of DSTs in seawater but had no distinct effect on DSTs content in the gill tissue. The decrease of OA content in the digestive gland and the increase of OA content in the culture medium, together with no significant change in OA content in the gill tissue gave some evidence that CA might not change the distribution of DSTs in mussel but facilitate the elimination of DSTs from the digestive gland. As for the OA in sea water, some studies have shown that OA in P. lima was compartmentalized in chloroplasts or vacuoles (Zhou and Fritz, 1994; Barbier et al., 1999; Lawrence and Cembella, 1999). So, the OA in sea water came mainly from mussels.
Zhou, J.; Fritz, L. Okadaic acid antibody localizes to chloroplasts in the DSP-toxin-producing dinoflagellates Prorocentrum lima and Prorocentrum maculosum. Phycologia 1994, 33, 455–461.
Barbier, M.; Amzil, Z.; Mondeguer, F.; Bhaud, Y.; Soyer-Gobillard, M.O.; Lassus, P. Okadaic acid and PP2A cellular immunolocalization in Prorocentrum lima (Dinophyceae). Phycologia 1999, 38, 41–46.
Lawrence, J.E.; Cembella, A.D. An immunolabeling technique for the detection of diarrhetic shellfish toxins in individual dinoflagellate cells. Phycologia 1999, 38, 60–65.
Point 2: DSTs were extracted with 80% MeOH from mussels. Was DTX3 extracted fully?
Response 2: Thank you very much for your professional comment. DSTs were extracted with 80% MeOH from mussels according to the manufacturer’s instructions of Okadaic Acid (DSP) ELISA Test Kit. We didn’t describe it clearly in the text. In order to ensure the extraction efficiency, the DSTs were extracted with 80% methanol twice successively. We are sorry that we could not guarantee the extraction efficiency of DTX-3. However, in our experiment, the DSTs-producing dinoflagellate P. lima (CCMP 2579) was purchased from the National Center for Marine Algae and Microbiota (NCMA), which has been proved to produce OA and DTX1 mainly. So, we think it probably does not affect our main findings.
Round 2
Reviewer 1 Report
Dear authors,
Thanks a lot for your revisions. Except my mistake, You just missed to cite (or at least answer) the paper I have suggested and which was published in Marine Drugs regarding the influence of biotic and abiotic factors. Even in experimental conditions, the conditions are not fully controlled du to the multitude of invisible biotic factors ...
Anyhow the paper was just a suggestion And I am pleased to let you know that I have now accepted your paper. Congrats!
Best,
The Reviewer
